# Nurses' occupational fatigue level and risk factors: A systematic review and meta-analysis

**Rong Pi**[1,2☯], **Yunfang Liu**[1☯], **Rong Yan**[1], **Zong De**[3], **Yali Wan**[1], **Yi Chen**[4], **Zihan He**[1,2], **Fang Liu**[1,2], **Yan Wang**[5]*, **Suyun Li**[1]*

**1** Department of Nursing, Union Hospital, Tongji Medical College, Huazhong University of Science and Technology, Wuhan, Hubei, China, **2** School of Nursing, Tongji Medical College, Huazhong University of Science and Technology, Wuhan, Hubei, China, **3** Department of Cardiology, Lhasa People's Hospital, Lhasa, Xizang, China, **4** Department of Clinical Nutrition, Wuhan Hospital of Traditional Chinese and Western Medicine (Wuhan No.1 Hospital), Tongji Medical College, Huazhong University of Science and Technology, Wuhan Hubei, China, **5** Department of Nursing, Jingshan union hospital of Huazhong University of Science and Technology, Jingshan, Hubei, China

☯ These authors contributed equally to this work.
* 616581645@qq.com (YW); lisuyun0503@163.com (SYL)

## Abstract

### Background

Occupational fatigue, characterized by both physical and mental exhaustion, is a pressing concern in various professions, particularly among nurses. Studies have consistently linked occupational fatigue to decreased productivity, heightened error rates, and compromised decision-making abilities, posing significant risks to both individual nurses and healthcare organizations. Despite its recognized impact, the global prevalence of occupational fatigue among nurses remains incompletely understood, and rigorous evaluations of the multifaceted factors influencing nurses' occupational fatigue are scarce in the existing literature.

### Objectives

This study aims to estimate the pooled score of nurses' occupational fatigue, and to systematically review the factors associated with nurses' occupational fatigue.

### Methods

The review searched eight databases, including PubMed, Web of Science, Scopus, CINAHL, PsycINFO, and Chinese databases such as China National Knowledge Infrastructure (CNKI), Chinese Biological Medical (CBM), and Wan Fang Database. The temporal scope of our search covers the period from the inception of each database to October 1, 2023. The PRISMA guidelines were followed in the reporting of the meta-analysis and systematic review. The meta-analysis was conducted using Stata 18.0 software, employing a random-effects model to pool the mean score and

**Data availability statement:** All relevant data are within the paper and its Supporting Information files.

**Funding:** This work was financially supported by Research on Humanities and Social Sciences by the Ministry of Education (21YJA630049). "The funder of Professor Suyun Li had designed the study and proofread the manuscript".

**Competing interests:** The authors have declared that no competing interests exist.

standard deviation of the Occupational Fatigue Exhaustion Recovery (OFER) scale. The restricted maximum-likelihood estimator was utilized to calculate the heterogeneity variance $\tau^2$. Subgroup analysis was conducted to explore sources of heterogeneity. The registration PROSPERO number is CRD42023456337.

## Results

After a rigorous selection process, 28 articles were ultimately deemed suitable for inclusion in this article, which encompassed a total of 13,290 registered nurses. The pooled mean scores for chronic fatigue, acute fatigue, and inter-shift recovery were 53.24 (95% CI: 48.42–58.28), 64.00 (95% CI: 60.62–67.38), and 47.37 (95% CI: 43.24–51.50), respectively. The subgroup analyses of occupational fatigue among nurses yield crucial insights into the way regional, age, temporal, and departmental factors interact to influence fatigue levels. In addition, the factors affecting nurses' occupational fatigue were found to include cultural, organizational and individual variables.

## Conclusion

A moderately high level of acute and chronic fatigue was observed among nurses, while the level of inter-shift recovery was low to moderate. It is imperative that healthcare systems provide enhanced support for nurses, a multifaceted approach is required, encompassing cultural shifts to reduce the normalization of overwork, organizational reforms to enhance staffing and scheduling, the implementation of fair compensation mechanisms, the development of targeted educational programs and individual-level interventions to promote healthier lifestyle practices.

## Introduction

The nursing profession is inherently demanding, characterized by long hours, high patient-to-nurse ratios, and emotionally taxing situations [1]. These factors contribute to an increased risk of both chronic and acute fatigue, which can have adverse effects not only on individual nurses but also on the overall healthcare system [2]. Research has demonstrated that occupational fatigue can impair cognitive function, reduce vigilance, and increase the likelihood of errors, thereby compromising patient safety and care outcomes [3–5]. This highlights the urgent need for comprehensive research into occupational fatigue, as the implications extend beyond individual health to encompass broader systemic concerns [6].

A substantial number of studies have been conducted to examine the prevalence and risk factors associated with occupational fatigue among nurses [7,8]. Previous studies have identified several key contributors to fatigue, including excessive workloads [9], inadequate staffing [10], shift work [11], and the emotional demands of patient care [12]. For example, studies have demonstrated that nurses who work night shifts or extended hours experience elevated levels of fatigue and burnout in

comparison to their daytime counterparts [13]. Furthermore, organizational culture plays a pivotal role, with work environments that prioritize productivity over employee well-being intensifying feelings of exhaustion and disengagement [14]. Nevertheless, while these studies offer valuable insights, they frequently concentrate on isolated factors or rely on disparate methodologies, resulting in a fragmented comprehension of the complex nature of occupational fatigue.

Notwithstanding the considerable body of research, a systematic synthesis of these findings through meta-analysis has yet to be conducted. The lack of rigorous meta-analytical studies limits the ability to draw comprehensive conclusions about the prevalence and determinants of occupational fatigue among nurses. Such a synthesis is essential for identifying patterns and interactions among various risk factors, which could inform targeted interventions [15]. Furthermore, many existing studies fail to consider the cultural and organizational contexts in which nursing occurs [16]. Factors such as institutional policies, leadership styles, and workplace environments can significantly influence fatigue levels, yet these elements are often overlooked in individual studies.

The necessity of addressing occupational fatigue is further underscored by the evolving landscape of healthcare [5]. As healthcare demands increase and the complexity of patient needs grows, the pressures on nurses are likely to intensify [17]. This necessitates a proactive approach to understanding and mitigating the factors contributing to occupational fatigue. Without appropriate interventions, the normalization of overwork may lead to a cycle of chronic fatigue and burnout, further exacerbating the nursing shortage and compromising patient care quality [18].

The objective of this study is to address the critical gaps by conducting a systematic review and meta-analysis. By synthesizing data from multiple databases, identifying the level of occupational fatigue among nurses and associated risk factors. Furthermore, subgroup analyses will be conducted to gain insights into how regional, age, temporal, and departmental factors interact to shape fatigue levels among nurses. The ultimate significance of this research lies in its potential to inform healthcare leaders and policymakers about effective strategies to combat occupational fatigue. By highlighting the multifaceted nature of fatigue and its determinants, we aim to advocate for systemic changes that promote nurse well-being and improve patient outcomes. This study not only seeks to advance our understanding of occupational fatigue in nursing but also emphasizes the urgent need for comprehensive, evidence-based interventions that address the root causes of fatigue within healthcare settings.

## Methods

### Design

This systematic review and meta-analysis were prospectively registered with PROSPERO (CRD42023456337). Our study was conducted according to the Preferred Reporting Items for Systematic Reviews and Meta-Analyses (PRISMA) guidelines and the Meta-analysis and Systematic Reviews of Observational Studies guidelines [19]. This study aims to quantify the level of occupational fatigue among registered nurses by estimating the pooled score of the Occupational Fatigue Exhaustion Recovery scale (OFER), and to systematically review the factors associated with nurses' occupational fatigue.

### Search strategy and selection criteria

The first and second authors systematically searched the databases. The search was conducted using eight databases: PubMed, Web of Science, Scopus, CINAHL, PsycINFO, and Chinese databases, including China National Knowledge Infrastructure (CNKI), Chinese Biological Medical (CBM), and Wan Fang Database. The search spanned the inception of the databases until October 1st, 2023. Given the language proficiency of our research team, we focused solely on studies published in English and Chinese.

Initial key words included "nurses", "nursing personnel", "registered nurses", "work fatigue", "occupational fatigue", "workplace fatigue", "tiredness", "chronic fatigue", "acute fatigue", "factors", "risk factors", and "dangerous factors". To ensure comprehensive coverage of the literature search, references cited in the literature were manually searched to

identify any relevant research not initially identified. We will also seek the assistance of an experienced librarian to refine the search strategy for each database. In cases where the full text is unavailable or only abstracts or unpublished documents are available, we will contact the corresponding author or first author for assistance. The full search strategy is presented in S1 Table. The inclusion and exclusion criteria are shown in Table 1.

### Study selection

To conduct an initial screening, all search results were imported into a reference management software (EndNoteX9). The first and second author then proceeded to screen the title and abstracts independently, utilizing a consensus-based screening process. EndNote was employed to organize the articles and assist with the removal of duplicates. In instances where initial agreement regarding article inclusion or exclusion was not reached, the full text of the articles in question was referred to by all authors to facilitate discussion and resolution of any discrepancies.

### Data extractionhe

Two authors independently extracted the data. Main data were extracted in Microsoft Office Excel, including: first author, country, aim, study design, service areas, sampling method, main finding, total sample, number of female and male, age, and sample size, mean, standard deviation of subscales (chronic fatigue, acute fatigue and inter-shift recovery) [20].

### Quality appraisal

As all studies included in this review are cross-sectional, the Agency for Healthcare Research and Quality (AHRQ) was used to assess methodological quality [21]. The AHRQ is currently regarded as an excellent tool for assessing the quality of cross-sectional studies. It comprises 11 items, including study design, participants, variables, data, and bias. A score of zero was assigned to responses indicating "No" or "Unclear," while a score of one was assigned to responses indicating "Yes." The AHRQ scores were categorized into three quality tiers: low (0–3), medium (4–7), and high (8–11). Poor quality suggested a higher probability of bias. The quality of the evidence was evaluated by R. P. and ZH. H. In the event of a disagreement, the matter will be resolved through discussion or consultation with a third party, YL. W. We also scrutinized the study reports for completeness, looking for any indications of unreported or selectively reported outcomes.

### Meta-analysis

The studies included in the analysis employed a consistent measurement instrument, thus enabling the use of meta-analysis to synthesize the quantitative data. The mean scores and standard deviations of the OFER subscale scores were

**Table 1. Inclusion and exclusion criteria for the systematic review.**

| Items | Inclusion | Exclusion |
|---|---|---|
| Study design | Observational studies | Qualitative studies/ experimental studies |
| Population | Registered nurses | Nurse managers, auxiliary nurses, and nurse educators |
| Instrument | The Occupational Fatigue Exhaustion Recovery scale (OFER) | --- |
| Outcomes | The sample size, mean, and standard deviation of the scale, and factors associated with occupational fatigue | insufficient data |
| Type of publication | Peer reviewed full text papers | Case reports, review articles, conference abstracts, comments, letters to the editor and protocols |
| Language | English or Chinese | Non-English and Non-Chinese |

aggregated across studies using Stata18 software. The pooled mean scores were then expressed as weighted effect sizes and 95% confidence intervals (CI). As anticipated, considerable between-study heterogeneity was observed; therefore, a random-effects model was employed to pool effect sizes. The restricted maximum-likelihood estimator was utilized to calculate the heterogeneity variance $\tau^2$ [22]. Subgroup analysis was conducted based on regions, age, years, and department in this study. To enhance the certainty of our findings, we conducted a thorough assessment of the robustness of the meta-analysis results. A sensitivity analysis was completed to ascertain whether any of the studies in the meta-analysis produced changes in outcome. Additionally, publication bias was assessed using Begg's tests. Regarding publication bias, in addition to using Begg's test, we also conducted a visual inspection of funnel plots to assess for asymmetry, which can be indicative of publication bias. Any asymmetry observed was further investigated by conducting trim-and-fill analyses to estimate the potential impact of missing studies on the overall effect estimate.

## Results

### Study selection

A total of 3288 studies were initially identified, of which 3259 studies were obtained through database searching and 23 studies were obtained from citation searching. Following the removal of duplicates, 2461 studies were retained for further consideration. Subsequently, 2438 studies were excluded through screening of titles and abstracts due to not fulfilling the inclusion criteria. Furthermore, 90 studies were omitted through screening of full texts. Finally, 28 studies (19 studies from the database and 9 from the citation search) met the eligibility criteria and were included in the review. (Fig 1)

### Study characteristics

Table 2 provides an overview of the main characteristics of the 28 included studies, which encompass a total of 13,290 registered nurses. The studies were published between 2006 and 2023. All studies were cross-sectional in design. The studies originated from a diverse range of countries across the global world, with the majority originating from the United States (n = 12; 42.86%), China (n = 5; 17.86%), South Korea (n = 4; 14.29%), and Saudi Arabia (n = 2; 7.14%). Additionally, one study was conducted in each of the following countries: Japan, Turkey, Jordan, New Zealand and Lebanon. The AHRQ scores ranged from 5 to 9. Fourteen studies were deemed to be of high quality, while the remaining fourteen studies were classified as middle quality. The risk of bias for the included studies was primarily attributable to item 2, item 6, and item 10. Details of the quality assessment for each study are presented in S2 Table.

### Nurses' occupational fatigue

Table 3 illustrates the mean scores for chronic fatigue, acute fatigue and inter-shift recovery as reported in the included studies. According to Winwood and colleagues, a score of 1–25 indicates a low level of construct for each subscale, while a score of 26–50 indicates a low to moderate level. A score of 51–75 suggests a moderate to high level, and a score of 76–100 indicates a high level[20]. The pooled estimates for chronic fatigue, acute fatigue and inter-shift recovery mean were 53.24 (95% CI: 48.42–58.28), 64.00 (95% CI: 60.62–67.38) and 47.37 (95% CI: 43.24–51.50), respectively (Figs 2–4). In accordance with the evaluation criteria of the OFER scale, the levels of chronic fatigue and acute fatigue among nurses were classified as moderate-high, while inter-shift recovery was classified as moderate-low. Significant heterogeneity was observed in the scores, and random-effects models were employed to pool the effect sizes: $I^2 = 99.37\%$, p = 0.00; $I^2 = 98.86\%$, p = 0.00; $I^2 = 99.26\%$, P = 0.00, respectively.

### Subgroup analyses of occupational fatigue of nurses

Subgroup analyses were conducted to examine the mean scores among chronic fatigue, acute and inter-shift recovery. The results of these analyses are presented in Table 4.

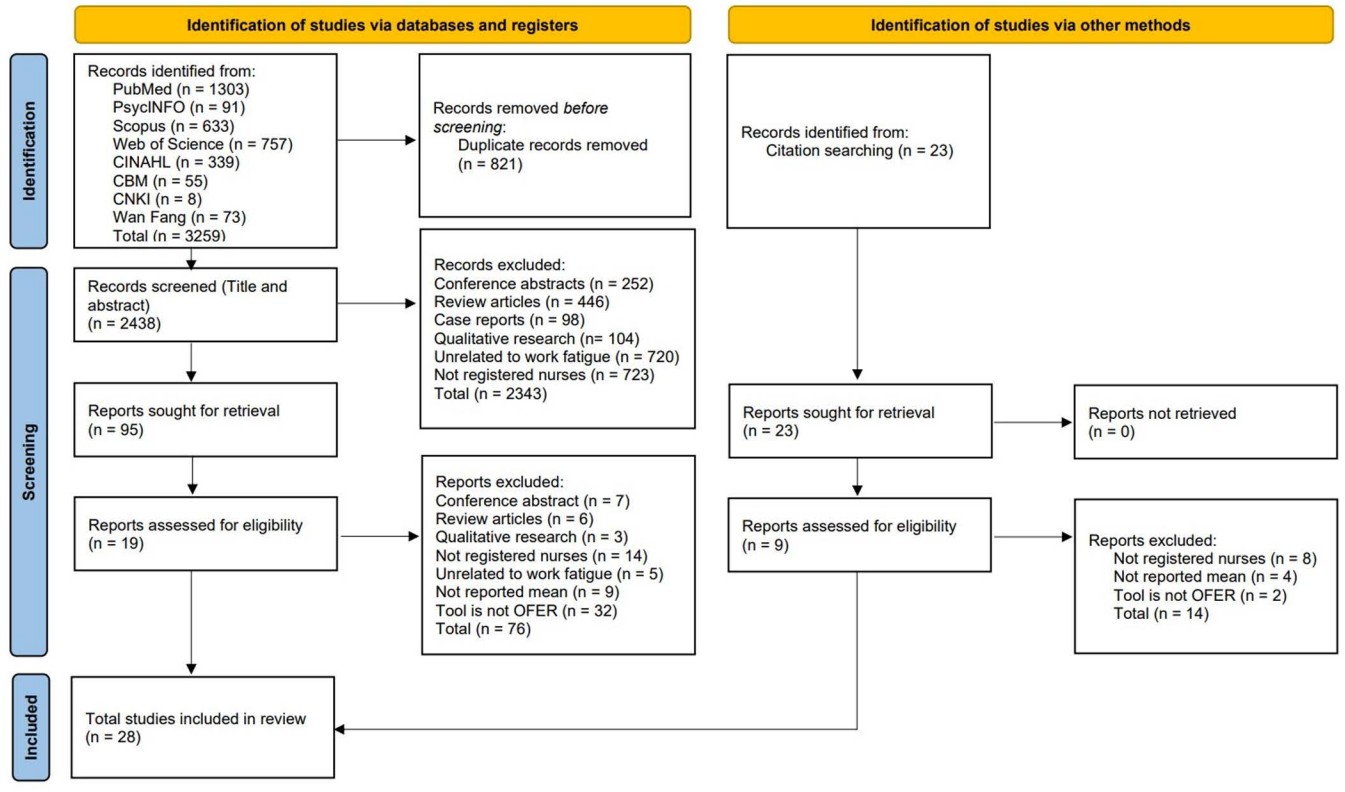

**Fig 1. The flow chart of the literature screening process and results.**

## Region

Country classifications were done based on the geographic regions. 9 studies were conducted in East Asia, 11 in America, and 5 in the Middle East. The results of the meta-analysis of the random effects model indicated that the Middle East exhibited the highest mean score for chronic fatigue (60.99), followed by East Asia (58.27) and America (47.82). The highest mean score for acute fatigue was observed in America (72.93), followed by the Middle East (65.78) and East Asia (63.83). Similarly, the Middle East exhibited the highest mean score for inter-shift recovery (56.00), followed by America (42.09) and East Asia (41.78).

## Age

There were 9 studies with a mean age of 20~29.99 years, 9 studies with a mean age of 30~39.99 years, and 4 studies with a mean age of 40~49.99 years. The results of the meta-analysis of the random effects model showed that the mean age of 40~49.99 had the lowest mean scores for inter-shift recovery (33.98) and chronic fatigue (47.99), while the highest mean score for acute fatigue (77.21). The mean age of 20~29.99 had the highest chronic fatigue (59.83) and inter-shift recovery (45.02) mean score.

## Years

8 studies were conducted in 2022 and 2023, 8 in 2019 and 2021, and 10 in 2009–2017. The results of the random effects model showed that studies conducted in 2019 and 2021 had the highest mean score for chronic fatigue (66.40) and the

Table 2. Characteristics of included studies.

| First author | Year | Country | Aim | Study design | Service areas | Sampling method | Main findings | Total sample | Quality assessment |
|---|---|---|---|---|---|---|---|---|---|
| Chen [23] | 2023 | China | To investigate the preventive behavior of occupational low back pain in junior nurses and analyze its influencing factors | Cross-sectional study | Hospital | Convenience sampling | The preventive behavior of occupational low back pain in junior nurses is at a medium level, nursing managers should take effective intervention measures according to the influencing factor, so as to improve their preventive and protective behavior level | 446 | High |
| Yamaguchi [24] | 2023 | Japan | To explore the association between fatigue and recovery and factors associated with recovery and chronic fatigue among nurses working a three-shift (8 hour shifts) or two-shift (more than 12 hour shifts) rotations in Japan | Cross-sectional study | Hospital | Convenience sampling | The three-shift rotations influenced nurses' inter-shift recovery more than the two-shift rotations. Regardless of shift patterns, managers must restrict overtime and encourage nurses to maintain sleep quality, family roles, and leisure activities. Moreover, considering nurses' age while selecting and organizing shift patterns may prevent chronic fatigue | 807 | High |
| Cho [25] | 2022 | The United States | to evaluate the relationships among nurse fatigue, individualized nursing care and nurse-reported quality of care | Cross-sectional study | Hospital | Convenience and snowball sampling | Nurses' higher levels of acute fatigue and chronic fatigue were associated with decreased perceptions of individualized nursing activities provided to patients on their last shifts | 858 | High |
| Min [26] | 2022 | South Korea | To determine the effects of work schedule characteristics on occupational fatigue and recovery among rotating-shift nurses in South Korea | Cross-sectional study | Acute care hospital | Convenience sampling | Overtime hours, number of night shifts, number of consecutive days off, and breaks were significant influential factors in some quantiles of acute fatigue, chronic fatigue, and inter-shift recovery, while total working hours was only associated with chronic fatigue in the 25th quantile | 436 | Middle |
| Cho [9] | 2022 | The United States | To evaluate the relationships between workload, nursing teamwork and nurse fatigue and the moderating effect of nursing teamwork on the relationship between workload and fatigue | Cross-sectional study | Hospital | Convenience and snowball sampling | All the nursing teamwork subscales (i.e., trust, team orientation, backup, shared mental model, team leadership) were significantly negatively related to acute and chronic fatigue. | 810 | High |
| Sagherian [27] | 2022 | The United States | To determine the long-term impact of the COVID-19 pandemic after 18 months on hospital nurses' insomnia, fatigue, burnout, post-traumatic stress, and psychological distress | Cross-sectional study | Hospital | Nonprobability convenience sampling | Nurses had subthreshold insomnia, moderate-to-high chronic fatigue, high acute fatigue, and low-to-moderate inter-shift recovery. Factors such as nursing experience, shift length, and frequency of rest breaks were significantly related to all well-being indices | 2488 | High |
| Alsayed [28] | 2022 | Saudi Arabia | To assess occupational fatigue "acute fatigue, chronic fatigue, and inter-shift recovery" and associated factors among Saudi nurses working 8-h shifts | Cross-sectional study | Public hospital | Convenience sampling | Saudi nurses rated themselves moderately fatigued with working 8-h shifts. Sleeping problems, meals per day, and frequency of exercise showed significant relations with chronic fatigue among nurses | 282 | High |

*(Continued)*

| First author | Year | Country | Aim | Study design | Service areas | Sampling method | Main findings | Total sample | Quality assessment |
|---|---|---|---|---|---|---|---|---|---|
| Qian [29] | 2022 | China | To explore the status and influencing factors of occupational fatigue of nurses in operating rooms in China | Cross-sectional study | Operating rooms | Convenience sampling | Job demand, job control, daily overtime hours, monthly total income, daily sleep time, social support, marital status, position and region accounted for 32.4% of the variation of acute fatigue. Job demand, job control, daily overtime hours, region and social support accounted for 24.2% of the variation of chronic fatigue | 1122 | High |
| Alshammari [12] | 2022 | Saudi Arabia | To explore the levels of fatigue and its associated factors among emergency department (ED) nurses in Saudi Arabia | Cross-sectional study | Emergency hospital | Convenience sampling | the Saudi Arabian ED nurses have high acute fatigue, moderate-high chronic fatigue, and, a high inter-shift recovery index. | 125 | Middle |
| Bolkan Günaydın [30] | 2022 | Turkey | To evaluate the pain, occupational fatigue, sleep, and quality of life in nurses and the relationships between them | Cross-sectional study | Hospital | Convenience sampling | The participants' average acute and chronic fatigues were at moderate-high fatigue levels. Musculoskeletal problems were common in nurses. Poor sleep hygiene and high chronic fatigue are significant risk factors. | 102 | Middle |
| Ross [31] | 2021 | The United States | To examine the levels, types, and factors associated with fatigue in registered nurses (RNs) in direct patient care (DCRNs) and in non-direct patient care (non-DCRNs) roles | Cross-sectional study | Hospital | Convenience sampling | DCRNs experienced higher total levels of acute and chronic fatigue than non-DCRNs, Inter-shift recovery may be particularly important in alleviating acute and chronic fatigue in nurses. | 313 | Middle |
| Hong [11] | 2021 | South Korea | To compare the fatigue, quality of life, turnover intention, and safety incident frequency between 2- and 3-shift nurses, and analyze their perceptions of the 2-shift system | Cross-sectional study | Hospital | Convenience sampling | 2-shift nurses had lower chronic fatigue and higher recovery between shifts and quality of life scores than 3-shift nurses | 227 | Middle |
| Min [32] | 2021 | South Korea | To examine the effects of work environments and occupational fatigue on care left undone in rotating shift nurses, and to identify the indirect (mediation) effect of work environments on care left undone through nurses' occupational fatigue in South Korean acute care hospitals | Cross-sectional study | Acute care hospital | Convenience sampling | High levels of occupational fatigue and poor inter-shift recovery among nurses can lead to care left undone. Nurses' occupational fatigue mediates the effect of work environment on care left undone | 488 | High |
| Mollica [10] | 2021 | The United States | To determine if there was a relationship between patient acuity and perceived fatigue | Cross-sectional study | Hospital | Convenience sampling | Most nurses experience substantial fatigue, with high acuity patients having an overall greater impact. NPAs shall contain fewer high acuity patients than lower acuity patients. Additionally, assignments should contain no more than five patients to mitigate fatigue | 114 | High |
| Orique [33] | 2019 | The United States | To examine medical-surgical nurses' capacity and tendency to perceive cues indicating clinical deterioration and nursing characteristics influencing deterioration cue perception | Cross-sectional study | Medical-surgical department | Convenience sampling | A significant association was found between nurses' capacity and tendency to perceive deterioration cues.. Fatigue, education, and certification were not identified as characteristics as sociated with deterioration cue perception | 86 | High |

*(Continued)*

| First author | Year | Country | Aim | Study design | Service areas | Sampling method | Main findings | Total sample | Quality assessment |
|---|---|---|---|---|---|---|---|---|---|
| Ismail [34] | 2019 | Jordan | To assess the psychosocial correlates of work-related fatigue among Jordanian emergency department nurses. | Cross-sectional study | Emergency department | Convenience sampling | The psychosocial factors correlated with all types of work-related fatigue were quantitative demands, work-family conflict, sexual harassment, threats of violence, physical violence, and bullying | 220 | Middle |
| Yu [16] | 2019 | New Zealand | To assess 12-h shift Intensive Care Unit (ICU) nurses' fatigue and identify the associated demographic factors | Cross-sectional study | Intensive Care Unit | Convenience sampling | More than half of the 12-h shift ICU nurses studied in both hospitals had low to moderate fatigue levels. Age, number of family dependents, years of nursing experience, and frequency of exercise per week were identified as key factors associated with fatigue | 67 | Middle |
| Min [35] | 2019 | South Korea | To determine the validity and reliability of the OFER-K scale with nurses working in Korean hospitals | Cross-sectional study | Hospital | Convenience sampling | The OFER-K scale is a reliable and valid instrument for assessing chronic fatigue, acute fatigue, and inter-shift recovery in Korean nurses | 331 | High |
| Sagherian [36] | 2017 | Lebanon | To explored fatigue, work schedules, and perceptions of nursing performance for a sample of Lebanese bedside nurses | Cross-sectional study | Hospital | Convenience sampling | Fatigue has safety implications for nurses' practice that should be monitored by nursing management | 77 | Middle |
| Blouin [37] | 2016 | The United States | To assess fatigue levels, and associated factors, explore any differences between fatigue levels, and determine whether other factors were associated with fatigue | Cross-sectional study | Hospital | Convenience sampling | Nursing administrators and shared governance councils can address the factors contributing to work-related fatigue and negatively impacting nursing personnel's ability to rest and recuperate | 1023 | Middle |
| Drake [38] | 2016 | The United States | To interpret hospital nurses' Fatigue using latent profile analysis | Cross-sectional study | Hospital | Convenience sampling | Nurses in the high fatigue/low recovery profile had the lowest values for adaptation variables, and nurses in the low fatigue/high recovery group had the highest values. | 227 | Middle |
| Liu [39] | 2016 | China | To test the fit of the hypothesized model for new nurses' intent to leave and determine the extent to which personal characteristics, work conditions, and work-related fatigue predict intent to leave among new nurses | Cross-sectional study | Hospital and medical center | Convenience sampling | Work conditions only had an effect through work-related fatigue on new nurses' intent to leave | 162 | Middle |
| Zhou [40] | 2015 | China | To describe the nursing job characteristics and nursing occupational fatigue situation at general hospitals in Chengdu in China and explore the correlations between job characteristics and occupational fatigue | Cross-sectional study | Hospital | Systematic sampling | Nurses' acute fatigue level and chronic fatigue level were high. Job control, job demand, and amount of shiftwork were important predictors of both acute fatigue and chronic fatigue | 923 | High |
| Chen [41] | 2014 | The United States | To investigate the status of acute fatigue, chronic fatigue and inter shift recovery among 12-hour shift nurses and how they differed by organisational and individual factors | Cross-sectional study | Acute care hospital | Convenience sampling | Nurses experienced a moderate to high level of acute fatigue and moderate levels of chronic fatigue and inter-shift recovery. Lack of regular exercise and older age were associated with higher acute fatigue | 130 | High |

*(Continued)*

**Table 2.** (Continued)

| First author | Year | Country | Aim | Study design | Service areas | Sampling method | Main findings | Total sample | Quality assessment |
|---|---|---|---|---|---|---|---|---|---|
| Hazzard [42] | 2013 | The United States | To describe postanesthesia care unit (PACU) nurses' fatigue and link fatigue levels to work- and nonwork-related factors. | Cross-sectional study | Postanesthesia care unit | Convenience sampling | Fatigue reduction strategies may account for these results including processes to ensure breaks are taken, use of a flex shift nurse to prevent shift overruns, and reduction of the number of three consecutive 12-hour shifts | 20 | Middle |
| Geiger-Brown [13] | 2012 | The United States | To describe sleep, sleepiness, fatigue, and neurobehavioral performance over three consecutive 12-h shifts for hospital registered nurses. | Cross-sectional study | Hospital | Stratified Sampling | Nurses working successive 12-h work shifts achieve an inadequate amount of sleep between shifts to recover physically or cognitively, irrespective of whether they work the day or the night shift. Nurses experienced greater sleepiness by their third consecutive 12-h shift than during their first two shifts of work, and night nurses appeared to be particularly vulnerable to sleepiness by the end of their work shift compared to day nurses. Neurobehavioral testing during actual work conditions has rarely been performed in nurses. | 80 | High |
| Barker [43] | 2011 | The United States | To investigate relationships between dimensions of fatigue and performances and differences in fatigue across levels of several demographic and work environment variables | Cross-sectional study | Hospital | Convenience sampling | Fatigue levels were negatively correlated with performance, further supporting the role of fatigue in nurse performance. Work environment variables were strongly associated with differences in perceived levels of fatigue | 745 | Middle |
| Fang [44] | 2009 | China | To examine predicting factors of fatigue in Chinese nurses | Cross-sectional study | Hospital | Systematic random sampling | Inter-shift recovery was the most important variable in the explanation of acute fatigue, while acute fatigue was the most important variable in the explanation of chronic fatigue | 581 | Middle |

lowest for inter-shift recovery (40.43), while studies conducted in 2022 and 2023 had the highest acute fatigue mean score (71.51).

## Department

18 studies were included in general hospitals and 8 studies in non-general hospitals (such as specialist hospitals and departments). The results of the random effects model showed that nurses in general hospitals had higher mean scores for chronic fatigue (53.53) and acute fatigue (69.44), but lower mean scores for inter-shift recovery (43.11).

## Publication bias and sensitivity analysis

The results obtained from Begg's test demonstrated that there was no statistically significant publication bias. The results for chronic fatigue, acute fatigue, and inter-shift recovery were 1.10 with $p = 0.270$, 0.26 with $p = 0.797$ and 0.16 with $p = 0.870$. In the sensitivity analysis, excluding each study from the analysis did not significantly change the pooled mean scores (S1–S3 Figs).

**Table 3. Mean scores of chronic fatigue, chronic fatigue and inter-shift recovery of included studies.**

| First author | Year | Gender | | | Chronic fatigue | | | Acute fatigue | | | Inter-shift recovery | | |
|---|---|---|---|---|---|---|---|---|---|---|---|---|---|
| | | Female | Male | Age | Sample | M | SD | Sample | M | SD | Sample | M | SD |
| Chen | 2023 | 57 | 389 | 25.34±2.71 | 446 | 45.12 | 15.88 | 446 | 64.28 | 18.54 | 446 | 55.92 | 16.03 |
| Yamaguchi | 2023 | 700 | 107 | NR | 807 | 51.79 | 19.15 | 807 | 63.47 | 18.51 | 807 | 36.49 | 19.40 |
| Cho | 2022 | 801 | 56 | 38.88±11.43 | 858 | 65.19 | 23.18 | 858 | 77.69 | 17.19 | NR | NR | NR |
| Min | 2022 | 412 | 24 | 26.40±3.48 | 436 | 73.39 | 16.79 | 436 | 70.40 | 15.04 | 436 | 29.82 | 15.67 |
| Cho | 2022 | 759 | 50 | 37.90±11.30 | 810 | 64.37 | 23.34 | 810 | 77.59 | 16.88 | NR | NR | NR |
| Sagherian | 2022 | 2231 | 256 | 41.50±12.32 | 2413 | 67.40 | 24.04 | 2412 | 78.99 | 18.04 | 2412 | 31.76 | 20.77 |
| Alsayed | 2022 | 271 | 11 | 30.58±6.33 | 282 | 52.27 | 23.19 | 282 | 57.01 | 17.12 | 282 | 50.60 | 13.08 |
| Qian | 2022 | NR | NR | 31.41±7.35 | 1045 | 63.80 | 18.83 | 1045 | 54.16 | 23.14 | 1045 | 43.99 | 15.81 |
| Alshammari | 2022 | 99 | 26 | 27.2±2.30 | 125 | 72.66 | 21.36 | 125 | 80.43 | 15.12 | 125 | 77.49 | 13.80 |
| Bolkan Günaydın | 2022 | 67 | 35 | 39.20±9.90 | 102 | 52.90 | 25.30 | 102 | 62.80 | 20.10 | 102 | 49.80 | 19.60 |
| Ross | 2021 | 222 | 19 | 47.59±11.52 | 120 | 51.69 | 27.82 | NR | NR | NR | 119 | 40.84 | 23.63 |
| Hong | 2021 | 217 | 10 | 28.46±4.17 | 227 | 64.51 | 15.16 | 227 | 68.63 | 17.81 | 227 | 42.88 | 18.66 |
| Min | 2021 | 461 | 27 | 26.43±3.46 | 488 | 73.02 | 16.72 | 488 | 69.86 | 15.16 | 488 | 30.13 | 15.57 |
| Mollica | 2021 | NR | NR | 32.80±9.70 | 114 | 56.70 | 26.20 | 114 | 60.00 | 11.30 | 114 | 53.30 | 8.30 |
| Orique | 2019 | 80 | 6 | 37.04±9.26 | 86 | 41.47 | 26.57 | 86 | 57.24 | 22.42 | 86 | 40.67 | 20.89 |
| Ismail | 2019 | 80 | 140 | 28.54±3.78 | 220 | 57.18 | 17.41 | 220 | 61.63 | 27.17 | 220 | 56.25 | 17.39 |
| Yu | 2019 | 51 | 16 | 38.00±8.60 | 67 | 27.00 | 22.9 | 67 | 49.90 | 12.20 | 67 | 46.40 | 10.50 |
| Min | 2019 | 315 | 16 | 26.77±3.86 | 331 | 73.98 | 16.71 | 331 | 71.91 | 14.67 | 331 | 28.25 | 15.21 |
| Sagherian | 2017 | 58 | 19 | NR | 77 | 70.26 | 21.70 | 77 | 76.80 | 18.70 | 77 | 39.24 | 18.57 |
| Blouin | 2016 | 947 | 122 | NR | 807 | 40.94 | 26.27 | 807 | 63.12 | 23.98 | 807 | 51.24 | 23.24 |
| Drake | 2016 | 211 | 16 | 46.41±11.43 | 223 | 36.70 | 24.14 | 199 | 50.52 | 20.49 | 210 | 57.14 | 22.16 |
| Liu | 2016 | 156 | 6 | 22.97±1.29 | 162 | 58.19 | 18.04 | 162 | 57.24 | 18.52 | 162 | 47.88 | 13.39 |
| Zhou | 2015 | NR | NR | 29.73±6.91 | 855 | 46.85 | 23.46 | 855 | 60.41 | 22.17 | 855 | 47.05 | 21.61 |
| Chen | 2014 | 130 | 0 | 36.80±10.70 | 130 | 47.30 | 21.80 | 130 | 65.60 | 18.60 | 130 | 50.00 | 19.50 |
| Hazzard | 2013 | 16 | 4 | 43.13±9.45 | 20 | 35.70 | 17.20 | 20 | 66.50 | 19.30 | 20 | 52.00 | 18.60 |
| Geiger-Brown | 2012 | 80 | 0 | 37.20±10.40 | 80 | 31.50 | 20.30 | 80 | 52.10 | 21.30 | 80 | 60.10 | 19.50 |
| Barker | 2011 | 702 | 43 | NR | 874 | 50.07 | 27.74 | 874 | 65.55 | 22.06 | 874 | 50.10 | 23.61 |
| Fang | 2009 | 581 | 0 | 29.49±6.73 | 581 | 47.14 | 23.38 | 581 | 63.4 | 21.69 | 581 | 45.62 | 22.44 |

## Factors related to occupational fatigue in nursing

As shown in Table 5, three categories of factors are most related to occupational fatigue, which included cultural factors (9 studies), organizational factors (17 studies), and individual factors (14 studies).

## Discussion

To the best of our knowledge, this is the inaugural quantitative meta-analysis to examine nurses' level of occupational fatigue according to the three dimensions of the OFER scale. The mean scores for chronic fatigue and acute fatigue were found to be at a moderate to high level, while the mean score for inter-shift recovery was at a moderate to low level. Sub-group analyses were performed to further investigate the impact of region, age, years of experience, and department on nurses' occupational fatigue. Additionally, a comprehensive systematic review was conducted to identify associated risk factors for nurses' occupational fatigue. This approach was taken with the aim of providing a more nuanced understanding of this critical aspect of nursing practice.

These findings highlight significant concerns for nurse well-being and patient safety.

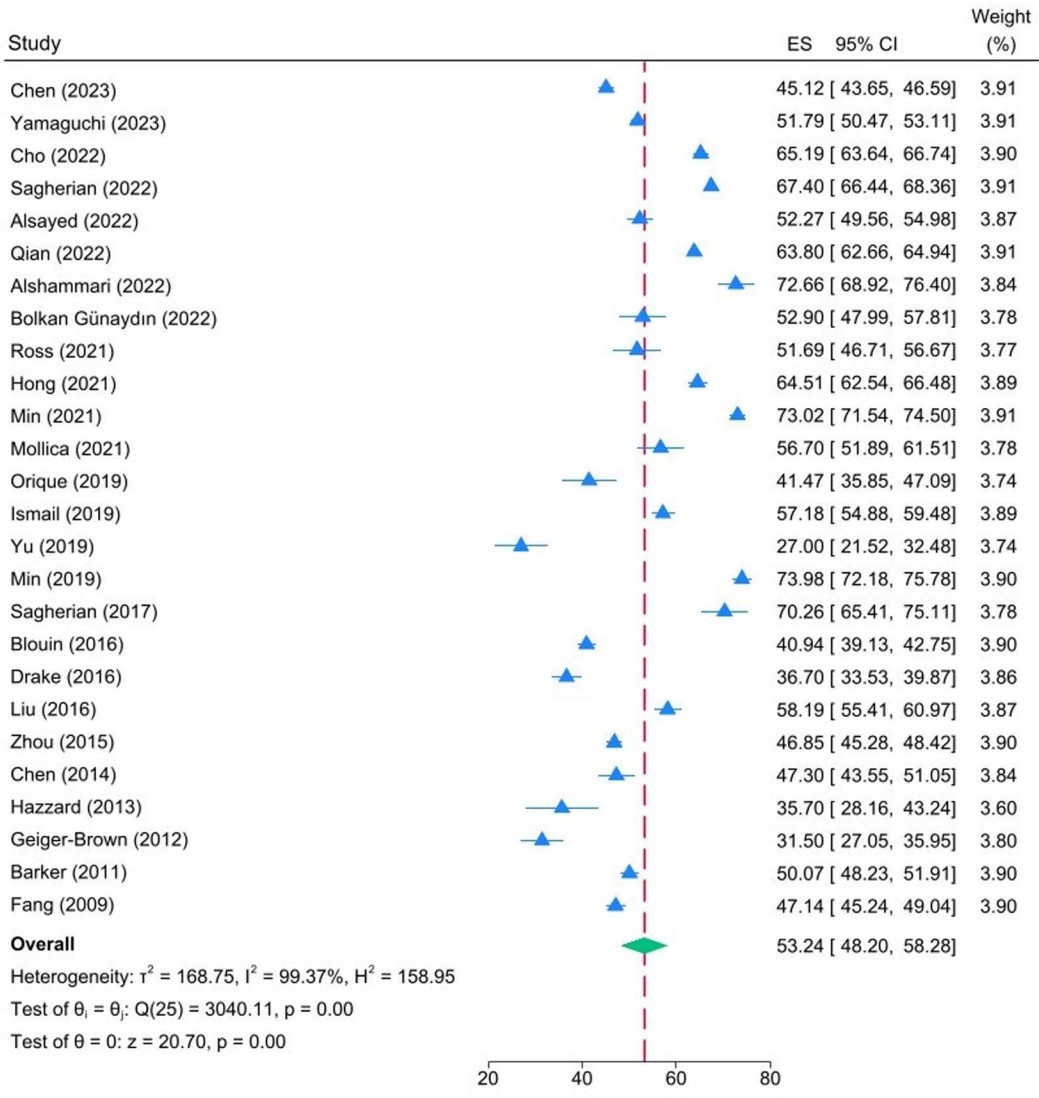

**Fig 2. Forest plot illustrating the pooled estimates for chronic fatigue mean.**

The moderate to high levels of chronic and acute fatigue observed in this study are consistent with the findings of previous research, which indicate that the demands of nursing (including long hours, high patient loads, and emotional strain) contribute to both sustained and short-term fatigue [25,27,45]. Chronic fatigue, which is caused by prolonged exposure to stressors, has been linked to burnout, reduced job satisfaction, and increased absenteeism [46]. The occurrence of acute fatigue within a shift is associated with a reduction in cognitive functioning and an elevated probability of medical errors, which collectively compromise patient safety [18]. The elevated levels of both types of fatigue highlight the necessity for systemic modifications, including enhanced staffing ratios, manageable workloads, and scientific shift scheduling.

The low levels of inter-shift recovery are a significant cause for concern, as inadequate recovery between shifts can exacerbate fatigue, impair sleep quality, and lead to long-term health risks such as cardiovascular issues [47]. It is probable that poor recovery is attributable to shift work schedules, frequent night shifts and an absence of sufficient time

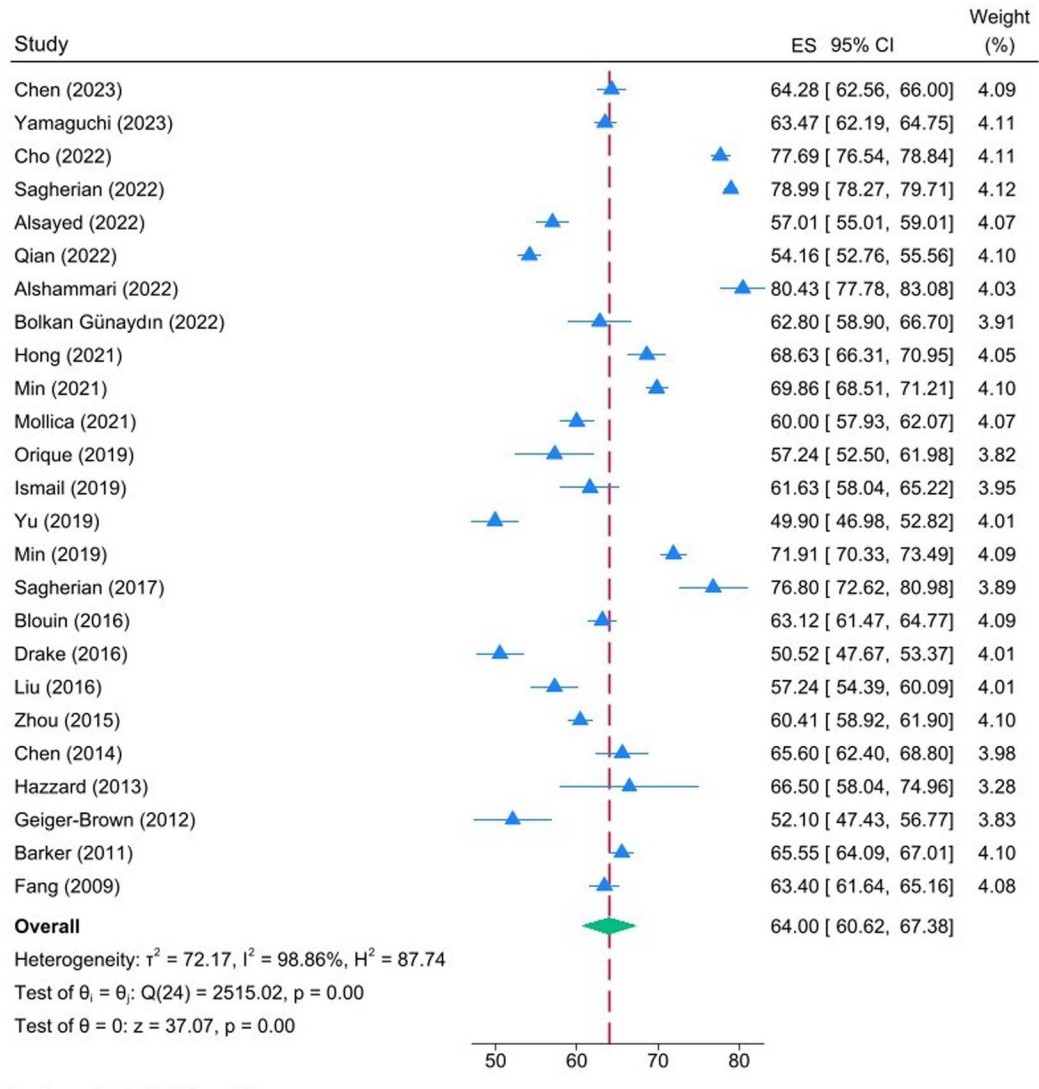

**Fig 3. Forest plot illustrating the pooled estimates for acute fatigue mean.**

between shifts, which disrupt circadian rhythms and impede physical and mental recuperation [48]. This emphasizes the necessity of implementing measures such as enhanced scheduling procedures, sufficient rest periods and mental health assistance to facilitate recuperation [49].

The subgroup analyses of occupational fatigue among nurses yield crucial insights into the way regional, age, temporal, and departmental factors interact to influence fatigue levels. The analysis indicates that nurses in the Middle East exhibit the highest mean score for chronic fatigue, which is significantly higher than the mean scores observed in other regions. This finding corroborates previous studies that have identified systemic challenges in this region, including high patient-to-nurse ratios and limited access to mental health resources [2,50]. Such environmental stressors can result in prolonged exposure to demanding work conditions, which in turn can lead to chronic fatigue. In contrast, the United States reported the highest acute fatigue, which corroborates the findings of previous studies. These findings suggest that high

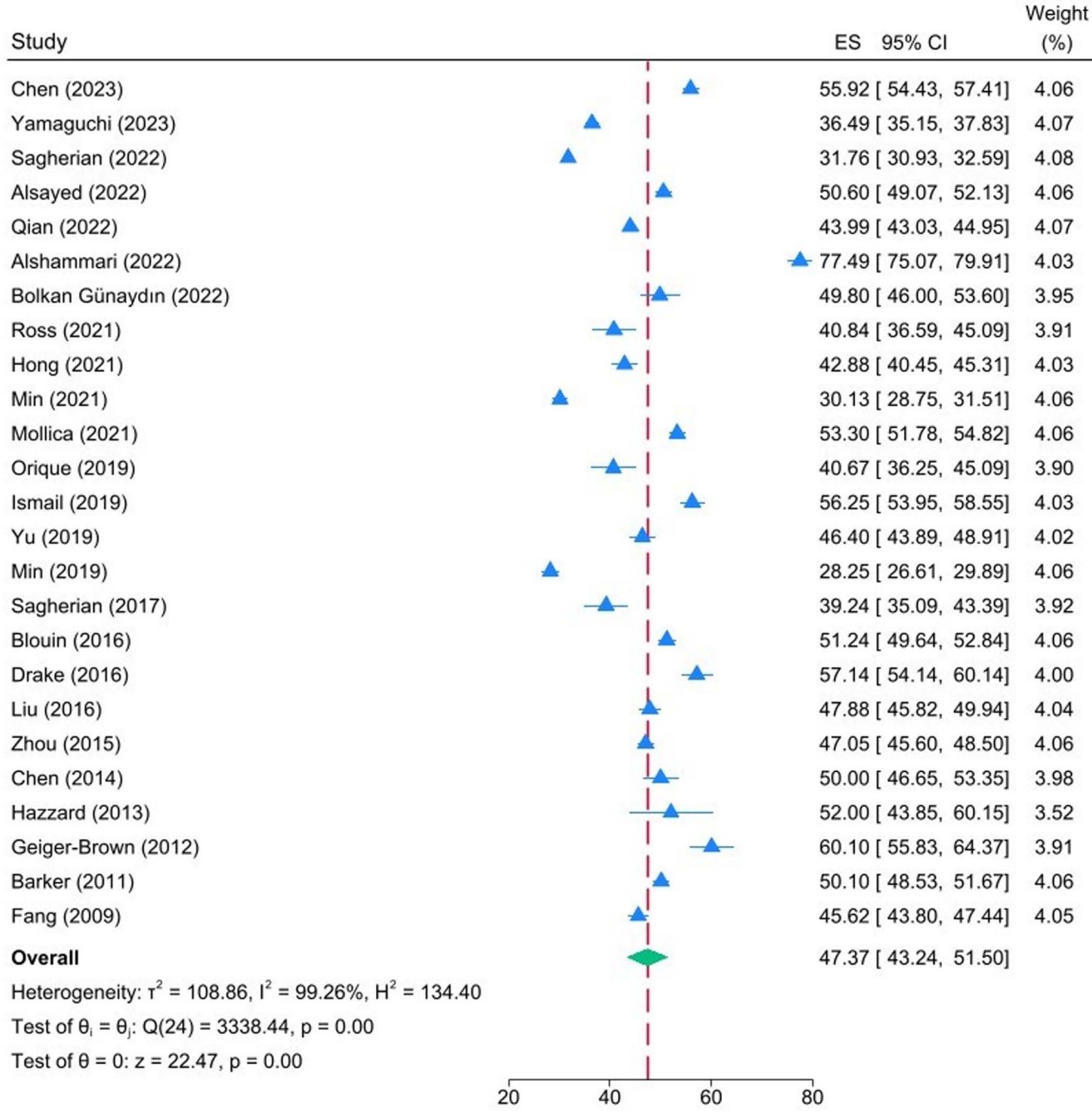

| Study | ES 95% CI | Weight (%) |
|---|---|---|
| Chen (2023) | 55.92 [ 54.43, 57.41] | 4.06 |
| Yamaguchi (2023) | 36.49 [ 35.15, 37.83] | 4.07 |
| Sagherian (2022) | 31.76 [ 30.93, 32.59] | 4.08 |
| Alsayed (2022) | 50.60 [ 49.07, 52.13] | 4.06 |
| Qian (2022) | 43.99 [ 43.03, 44.95] | 4.07 |
| Alshammari (2022) | 77.49 [ 75.07, 79.91] | 4.03 |
| Bolkan Günaydın (2022) | 49.80 [ 46.00, 53.60] | 3.95 |
| Ross (2021) | 40.84 [ 36.59, 45.09] | 3.91 |
| Hong (2021) | 42.88 [ 40.45, 45.31] | 4.03 |
| Min (2021) | 30.13 [ 28.75, 31.51] | 4.06 |
| Mollica (2021) | 53.30 [ 51.78, 54.82] | 4.06 |
| Orique (2019) | 40.67 [ 36.25, 45.09] | 3.90 |
| Ismail (2019) | 56.25 [ 53.95, 58.55] | 4.03 |
| Yu (2019) | 46.40 [ 43.89, 48.91] | 4.02 |
| Min (2019) | 28.25 [ 26.61, 29.89] | 4.06 |
| Sagherian (2017) | 39.24 [ 35.09, 43.39] | 3.92 |
| Blouin (2016) | 51.24 [ 49.64, 52.84] | 4.06 |
| Drake (2016) | 57.14 [ 54.14, 60.14] | 4.00 |
| Liu (2016) | 47.88 [ 45.82, 49.94] | 4.04 |
| Zhou (2015) | 47.05 [ 45.60, 48.50] | 4.06 |
| Chen (2014) | 50.00 [ 46.65, 53.35] | 3.98 |
| Hazzard (2013) | 52.00 [ 43.85, 60.15] | 3.52 |
| Geiger-Brown (2012) | 60.10 [ 55.83, 64.37] | 3.91 |
| Barker (2011) | 50.10 [ 48.53, 51.67] | 4.06 |
| Fang (2009) | 45.62 [ 43.80, 47.44] | 4.05 |
| **Overall** | 47.37 [ 43.24, 51.50] | |

Heterogeneity: $\tau^2 = 108.86$, $I^2 = 99.26\%$, $H^2 = 134.40$

Test of $\theta_i = \theta_j$: Q(24) = 3338.44, p = 0.00

Test of $\theta = 0$: z = 22.47, p = 0.00

Random-effects REML model

**Fig 4. Forest plot illustrating the pooled estimates for inter-shift recovery mean.**

turnover rates and intense workloads in American healthcare settings contribute to significant emotional and physical strain [51]. The findings indicate that younger nurses (aged 20–29.99 years) experience higher levels of chronic fatigue but demonstrate superior inter-shift recovery. This apparent contradiction can be elucidated through the lens of resilience

**Table 4. Subgroup analyses of mean scores among chronic fatigue, acute fatigue and inter-shift recovery.**

| Variable | Number of studies | Effect model | Chronic fatigue | | | | Acute fatigue | | | | Inter-shift recovery | | | |
|---|---|---|---|---|---|---|---|---|---|---|---|---|---|---|
| | | | sample size | mean score (95%CI) | I2 | P | sample size | mean score (95%CI) | I2 | P | sample size | mean score (95%CI) | I2 | P |
| **Region** | | | | | | | | | | | | | | |
| East Asia | 9 | random | 4942 | 58.27 (50.90, 65.63) | 99.5% | <0.001 | 4942 | 63.83 (63.29, 64.36) | 98.1% | <0.001 | 4942 | 41.78 (41.29, 42.27) | 99.2% | <0.001 |
| America | 11 | random | 5725 | 47.82 (39.77, 55.86) | 99.2% | <0.001 | 5580 | 72.93 (72.44, 73.43) | 99.2% | <0.001 | 4852 | 42.09 (41.52, 42.66) | 99.3% | <0.001 |
| Middle East | 5 | random | 806 | 60.99 (53.24, 68.74) | 99.2% | <0.001 | 806 | 65.78 (64.49, 67.08) | 98.2% | <0.001 | 806 | 56.00 (54.95, 57.04) | 99.1% | <0.001 |
| **Age** | | | | | | | | | | | | | | |
| 20~29.99 | 9 | random | 3435 | 59.83 (51.40, 68.26) | 99.4% | <0.001 | 3435 | 66.65 (66.03, 67.27) | 97.4% | <0.001 | 3435 | 45.02 (44.43, 45.61) | 99.6% | <0.001 |
| 30~39.99 | 9 | random | 2764 | 48.88 (41.62, 56.15) | 98.4% | <0.001 | 2764 | 64.12 (63.43, 64.80) | 99.1% | <0.001 | 1906 | 47.73 (47.08, 48.38) | 95.9% | <0.001 |
| 40~49.99 | 4 | random | 2776 | 47.99 (28.94, 67.04) | 99.3% | <0.001 | 2631 | 77.21 (76.51, 77.90) | 99.5% | <0.001 | 2761 | 33.98 (33.20, 34.76) | 99.0% | <0.001 |
| **Years** | | | | | | | | | | | | | | |
| 2022–2023 | 8 | random | 6078 | 60.31 (59.78, 60.84) | 99.3% | <0.001 | 6077 | 71.51 (71.04, 71.97) | 99.6% | <0.001 | 5219 | 42.18 (41.70, 42.67) | 99.7% | <0.001 |
| 2019–2021 | 8 | random | 1653 | 66.40 (65.54, 67.26) | 98.7% | <0.001 | 1533 | 66.61 (65.82, 67.40) | 97.7% | <0.001 | 1652 | 40.43 (39.71, 41.15) | 99.2% | <0.001 |
| 2009–2017 | 10 | random | 3809 | 46.70 (45.93, 47.47) | 97.0% | <0.001 | 3785 | 62.29 (61.60, 62.99) | 94.9% | <0.001 | 3796 | 49.09 (48.41, 49.77) | 91.6% | <0.001 |
| **Department** | | | | | | | | | | | | | | |
| General hospital | 18 | random | 9359 | 53.53 (48.26, 58.80) | 99.2% | <0.001 | 9214 | 69.44 (69.05, 69.82) | 99.1% | <0.001 | 8486 | 43.11 (42.70, 43.51) | 99.2% | <0.001 |
| Non general hospital | 8 | random | 2181 | 52.68 (44.65, 60.71) | 98.8% | <0.001 | 2181 | 63.12 (62.31, 63.92) | 98.6% | <0.001 | 2181 | 44.71 (44.05, 45.37) | 99.4% | <0.001 |

theory, which postulates that younger individuals may possess adaptive coping strategies that facilitate recovery [52]. In contrast, nurses aged 40–49.99 years exhibited the highest levels of acute fatigue and the lowest levels of recovery. This suggests that older healthcare workers often face cumulative stress and health challenges that can amplify fatigue. It is imperative that tailored interventions be developed and implemented for older nurses, with a particular focus on stress management and physical well-being.

The highest chronic fatigue scores during 2019–2021 correlate with the onset of the COVID-19 pandemic, previous studies also documented heightened stress levels among healthcare workers during this period [53]. The subsequent high acute fatigue in studies from 2022–2023 indicates that the effects of the pandemic are still being felt, as nurses continue

**Table 5. Factors related to nurses' occupational fatigue.**

| Categories | Detailed factors |
|---|---|
| Cultural factors | Magnet hospital, social support, workplace (job/ work) support, support of supervisor, nursing/unit teamwork, rust in management, justice, called to work on day off, safety culture, sexual harassment, threats of violence, bullying |
| Organizational factors | shift length, three-shift, number of night shift, number of consecutive days off, breaks, 30-min breaks, hours of sleep, (daily) over time hour, total working hours, workload, Job demand, number of patients per nurse, nurse-patient ratio, rewards, monthly total income |
| Individual factors | age, gender, years of experience, position, role clarity, Sleeping problems, refreshed after sleep, sleep disturbance, need for recovery, meals day, regular exercise(frequency of exercise per week), self-rated health, job control, emotional demand, influence at work, commitment at the workplace, stress, burnout, job strain, maladaptive coping, commitment to workplace, recognition, job satisfaction, depression, brief resilience, compassion satisfaction, marital status, family caregiving responsibility, work-family conflict, the number of dependents |

to face increased workloads and emotional strain. This emphasizes the necessity for the provision of ongoing mental health resources and organizational support to address the issue of sustained fatigue [54]. The nurses in general hospitals who participated in the study reported higher chronic fatigue and acute fatigue, as well as lower inter-shift recovery. These findings reflect the demanding nature of the work environment in general hospitals. Prior research has demonstrated that general hospital settings frequently have higher patient loads and more intricate care requirements, which can result in elevated fatigue levels. This highlights the imperative for systemic modifications, such as enhanced staffing ratios and augmented workplace assistance, to mitigate the burden on nurses in these settings.

This systematic review identifies three categories of risk factors for nurses' occupational fatigue: cultural, organizational, and individual factors. Cultural factors, such as poor nursing teamwork, inadequate workplace support, and limited social support, serve to exacerbate stress by weakening collaborative efforts and reducing emotional resilience [55]. Conversely, a supportive work culture, characterized by collaboration and strong interpersonal relationships, can act as a buffer against the effects of workplace stress, thus reducing fatigue [56]. Organizational factors, including high workloads, insufficient staffing, and disruptive shift schedules (especially night and rotating shifts), as well as salary and treatment, have been identified as key contributors to fatigue. These factors directly impact recovery time and disrupt circadian rhythms, leading to fatigue. A further intensification of fatigue is caused by a lack of control over schedules and poor management support [57]. To address these issues, it is necessary to implement systemic changes, including improvements in staffing ratios, optimization of shift scheduling, and the provision of greater managerial support and autonomy for nurses.

It is also evident that individual factors, such as age, health status, work-family balance and coping mechanisms, exert a significant influence on how nurses cope with fatigue. Those with poor sleep hygiene, inadequate stress management, or pre-existing health issues are more susceptible to fatigue, while younger and older nurses may encounter challenges due to experience or physical limitations [58]. To address these issues, systemic changes are required, including improvements in staffing ratios, optimization of shift scheduling, and the provision of greater managerial support and autonomy for nurses. A multifaceted approach is necessary to address occupational fatigue among nurses, targeting cultural, organizational, and individual risk factors. To reduce fatigue, systemic interventions are required, including the fostering of supportive team dynamics, improvements in staffing policies, and the promotion of personal well-being practices to ensure nurse health and enhance patient care quality.

In general, the results of the meta-analysis provide invaluable insights into the level of occupational fatigue among registered nurses. By incorporating relevant literature, this review reveals the cultural, organizational and individual

factors associated with nurses' occupational fatigue. The rigorous methodology employed in this systematic review and meta-analysis, including the use of predefined inclusion criteria, quality assessment of studies, and data statistical analysis, enhances the robustness and reliability of our findings.

The implications of these results are profound for nursing managers and healthcare organizations. The findings emphasize the critical need to address cultural, organizational, and individual factors contributing to nurses' occupational fatigue. To this end, strategies such as fostering a positive work environment, ensuring adequate staffing levels, promoting respect and collaboration among team members, and building psychological resilience among nurses are essential. Policy-makers, on the other hand, should enforce regulations aimed at reducing stressors in the workplace and ensuring a supportive environment for nurses. This includes implementing policies that promote work-life balance, provide adequate resources, and ensure compliance with best practices in nursing care. Future research should delve deeper into how these factors interplay in predicting fatigue and explore innovative interventions to mitigate its effects. By filling these knowledge gaps, we can better support nurses' health and professional development, ultimately leading to improved patient care and outcomes through evidence-based strategies.

## Limitations

It should be noted that the review is not without limitations. Firstly, we observed a high degree of heterogeneity, which could not be mitigated through subgroup analysis, thereby impacting the accuracy of our results. Consequently, further research is warranted. Secondly, the included studies employed a cross-sectional design. Consequently, it is not possible to infer a causal relationship. Longitudinal studies were required in future. Thirdly, the sample populations of the included studies included registered nurses, so the results may not be fully representative of the broader nursing workforce, particularly in diverse healthcare settings. Future research should ensure that samples are reflective of the broader nursing workforce, particularly in countries with diverse healthcare settings. This is especially important for developing targeted interventions aimed at improving nurses' work environments and well-being.

## Conclusion

A moderately high level of acute and chronic fatigue was observed among nurses, while the level of inter-shift recovery was low to moderate. Our findings indicate that nursing fatigue is a critical issue that requires both organizational and individual-level interventions. It is imperative that these concerns be addressed to maintain the well-being of nurses, reduce burnout, and improve the quality of patient care. It is recommended that future research be directed toward the development of solutions that enhance recovery and mitigate fatigue, with the objective of supporting a healthier nursing workforce. Through these efforts, we hope to contribute to the development of healthier work environments that prioritize the well-being of nurses and the quality of care they provide.

## Supporting information

**S1 Table. The full searching strategy.**
(DOCX)

**S2 Table. Quality assessment for studies.**
(DOCX)

**S3 Table. Included and excluded studies in the systematic review and meta-analysis.**
(XLSX)

**S1 Fig. Sensitivity analysis for chronic fatigue subscale.**
(TIFF)

 

**S2 Fig. Sensitivity analysis for acute fatigue subscale.**
(TIFF)

**S3 Fig. Sensitivity analysis for inter-shift subscale.**
(TIFF)

**S1 Checklist. PRISMA checklist1.17.docx. a PRISMA checklist.**
(DOCX)

## Acknowledgments

We would like to thank all the authors of our included studies.

## Author contributions

**Conceptualization:** Rong Pi, Yunfang Liu, Yan Wang.

**Data curation:** Rong Pi, Yunfang Liu, Zihan He, Fang Liu, Yan Wang, Suyun Li.

**Formal analysis:** Rong Pi, Yan Wang.

**Funding acquisition:** Suyun Li.

**Investigation:** Rong Pi, Yunfang Liu, Suyun Li.

**Methodology:** Rong Pi, Yunfang Liu, Rong Yan, Zong De, Yali Wan, Yi Chen, Zihan He, Fang Liu, Suyun Li.

**Project administration:** Suyun Li.

**Software:** Rong Pi, Rong Yan, Zong De, Yali Wan, Yi Chen, Zihan He, Fang Liu.

**Supervision:** Rong Yan, Zong De, Yali Wan, Yi Chen, Yan Wang, Suyun Li.

**Validation:** Yali Wan, Yan Wang, Suyun Li.

**Visualization:** Yan Wang, Suyun Li.

**Writing – original draft:** Rong Pi, Yunfang Liu.

**Writing – review & editing:** Rong Pi, Yi Chen, Yan Wang, Suyun Li.

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
