## [Decision Letter · Decision Letter 0]

PONE-D-24-45547Nurses’ occupational fatigue level and risk factors: a Meta analysis and Systematic ReviewPLOS ONE

Dear Dr. Li,

Thank you for submitting your manuscript to PLOS ONE. After careful consideration, we feel that it has merit but does not fully meet PLOS ONE’s publication criteria as it currently stands. Therefore, we invite you to submit a revised version of the manuscript that addresses the points raised during the review process.

We look forward to receiving your revised manuscript.

Kind regards,

Fatma Refaat Ahmed, Ph.D.

Academic Editor

PLOS ONE

2. Thank you for stating the following financial disclosure:  [This work was financially supported by Research on Humanities and Social Sciences by the Ministry of Education(21YJA630049).]. 

3. In the online submission form, you indicated that [The inquiries should be directed to the corresponding author, and data from this study will be made available upon reasonable request.re].

4. As required by our policy on Data Availability, please ensure your manuscript or supplementary information includes the following:

Reviewers' comments:

Reviewer's Responses to Questions

**Comments to the Author**

1. Is the manuscript technically sound, and do the data support the conclusions?

Reviewer #1: Yes

Reviewer #2: Partly

2. Has the statistical analysis been performed appropriately and rigorously? 

Reviewer #1: Yes

Reviewer #2: Yes

3. Have the authors made all data underlying the findings in their manuscript fully available?

Reviewer #1: Yes

Reviewer #2: Yes

4. Is the manuscript presented in an intelligible fashion and written in standard English?

Reviewer #1: Yes

Reviewer #2: Yes

5. Review Comments to the Author

Reviewer #1: REVIEWER SUGGETIONS AND COMMENT

Generally congratulation for the interesting paper which informed the good health of the Nurses due to their duties and responsibility. However minor issue should be improving.

Adhere to journals guideline

TITILE

When I read the title, I was wondering if it was a systematic review or a meta-analysis, Please be specific.

ABSTRACT

The authors should update and make it scientific sound when reading part of the background. Additionally, the purpose needs to be revised.

Method: The authors should explain the study period (start and finish dates). Design sample size and analysis used.

METHODS

On part of study design the authors should make it clear. Will make the confusion to the reader.

Reporting bias assessment: The author should mention any procedures used to assess the likelihood of bias in a synthesis due to missing results (caused by reporting biases).Also, identify any methodologies used to determine certainty (or confidence) in the body of evidence supporting an outcome.

RESULT

On study selection the authors should use the flow chart diagram

The authors should arrange the section of the result well, as the same figure and table are not seen. Review and improve.

Discussion

On part of limitations the authors should improve and make it clear

Discuss implications of the results for practice, policy, and future research.

References

Several references do not fit the requirements of Vancouver style. Revise and improve them

Reviewer #2: Dear Author,

Your study is very valuable, and I have some comments that may help improve your research:

1. The abstract does not clearly articulate the necessity of the work. Please make this more explicit.

2. Include a comprehensive list of your inclusion and exclusion criteria.

3. Specify the keywords used in your search and the types of studies that were included.

4. Were there no studies published before 2009?

5. Indicate how many authors contributed to the search and quality assessment of the studies.

6. In Table 1, please add the data collection instrument used for each study.

7. At the end of the discussion, summarize the strengths of your study and provide suggestions for future research.

Thank you for considering these points.

6. PLOS authors have the option to publish the peer review history of their article (what does this mean? ). If published, this will include your full peer review and any attached files.

**Do you want your identity to be public for this peer review?** For information about this choice, including consent withdrawal, please see our Privacy Policy .

Reviewer #1: No

Reviewer #2: No

---

## [Author Response · Author response to Decision Letter 1]

17 Jan 2025

Response to editor’s comments

Thank you for providing us with the reviewers' comments. We would like to inform you that each comment and suggestion has been addressed in detail.

Q1. Please ensure that your manuscript meets PLOS ONE's style requirements, including those for file naming.

Response: Thank you for your comments. The manuscript has been fully reviewed and revised to meet PLOS ONE's style requirements.

Q2. Please state what role the funders took in the study. If the funders had no role, please state: "The funders had no role in study design, data collection and analysis, decision to publish, or preparation of the manuscript." If this statement is not correct you must amend it as needed. Please include this amended Role of Funder statement in your cover letter; we will change the online submission form on your behalf.

Response: Thank you for your careful check. The role of funders has been stated. The funder of Professor Suyun Li had designed the study and proofread the manuscript.

Q3. All PLOS journals now require all data underlying the findings described in their manuscript to be freely available to other researchers, either 1. In a public repository, 2. Within the manuscript itself, or 3. Uploaded as supplementary information.

Response: We sincerely thank you for your careful reading. The data has been uploaded as supplementary information. As this study is a meta-analysis, the data used are exclusively from publicly available literature. These literatures have been cited in detail in the paper. We have ensured that all references are publicly available, thus ensuring the transparency and reproducibility of the study.

Q4. As required by our policy on Data Availability, please ensure your manuscript or supplementary information includes the following.

Response: We sincerely appreciate your valuable comment. We have uploaded the data as supplementary information. S2 Table shows the quality assessment for each study. S3 Table shows the list of the studies that were included and excluded in the systematic review and meta-analysis. There is also information included in the main text. Fig 1 shows the literature screening process and results, Table 2 shows characteristics of included studies, and Table 3 shows mean scores of chronic fatigue, chronic fatigue, and inter-shift recovery of included studies.

Q5. Please review your reference list to ensure that it is complete and correct.

Response: Thanks for your valuable comment. We have carefully reviewed the references and revised them.

Replies to Reviewer #1

Reviewer #1:

Q1. Generally, congratulation for the interesting paper which informed the good health of the Nurses due to their duties and responsibility. However minor issue should be improving. TITILE: When I read the title, I was wondering if it was a systematic review or a meta-analysis, Please be specific.

Response: Thank you for your comment. I would like to clarify the distinction between a meta-analysis and a systematic review and explain how they are utilized in our study.

A meta-analysis is a statistical technique used to combine the results of multiple independent studies to derive a more precise and reliable conclusion. It typically involves pooling the effect sizes from multiple studies and assessing the heterogeneity among these studies. A systematic review is a comprehensive, systematic method of reviewing the literature to identify, select, and synthesize all relevant evidence on a specific question or research area. A systematic review can include a meta-analysis or not. When a systematic review includes a meta-analysis, it is used to quantitatively synthesize the results of multiple studies. When it does not include a meta-analysis, the systematic review focuses on describing and synthesizing the study results.

In our study, we conducted both a meta-analysis and a systematic review. Specifically, for the section exploring the occupational fatigue level of clinical nurses, we conducted a meta-analysis to quantitatively synthesize the results of multiple studies and derive a more accurate conclusion. For the section exploring the factors influencing occupational fatigue among clinical nurses, due to the large number of factors and their difficulty in quantitative synthesis, we conducted a systematic review to describe and synthesize the results of multiple studies. To avoid misunderstandings and following the naming conventions of other literature, we note that a systematic review can include or exclude a meta-analysis.

Therefore, we adjusted the position of "systematic review" and "meta-analysis" in the title to more accurately reflect the content of our study. The new title is "Nurses' occupational fatigue level and risk factors: A Systematic Review and Meta-analysis." This title clearly indicates that our study includes both a systematic review and a meta-analysis, with the meta-analysis focusing on the occupational fatigue level of clinical nurses and the systematic review focusing on the factors influencing occupational fatigue.

Q2. ABSTRACT: The authors should update and make it scientific sound when reading part of the background. Additionally, the purpose needs to be revised.

Response: Thank you for your constructive feedback. Regarding your comment that the abstract's background section should be updated and made to sound more scientific, we have revised the background accordingly. We have provided a clearer description of occupational fatigue, its characteristics, and its impact on both individual nurses and healthcare organizations. Furthermore, we have highlighted the incomplete understanding of the global prevalence and severity of occupational fatigue among nurses, as well as the scarcity of rigorous evaluations of its associated factors. Please refer to the revised version on page 2, lines 23-30.

In response to your suggestion to revise the purpose, we have clarified our objectives. Our primary aim is now to estimate the pooled score of nurses' occupational fatigue, providing a comprehensive view of its prevalence. Additionally, we aim to systematically review and synthesize the evidence regarding the factors associated with nurses' occupational fatigue, shedding light on areas that require further investigation and intervention. Please refer to the revised version on page 2, lines 31-32.

Q3. Methods: The authors should explain the study period (start and finish dates). Design sample size and analysis used.

Response: Thank you very much for your thorough review and for providing valuable feedback. We have carefully considered your suggestions and have made corresponding revisions to the manuscript. We have explicitly stated the search period, which ranged from the inception of each database to October 1, 2023, to ensure readers have a clear understanding of the temporal scope of our study. Please refer to the revised version on page 2, lines 36-37.

As for the design sample size and analysis used, we have provided detailed information on the number of articles included in the study and the total sample size (13290 registered nurses). Please refer to the revised version on page 3, lines 44-45.

We have also described the analytical methods used, including the use of Stata 18.0 software for random-effects meta-analysis and the application of the restricted maximum likelihood estimator to calculate heterogeneity variance. Please refer to the revised version on page 2, lines 38-42.

Q4. METHODS: On part of study design the authors should make it clear. Will make the confusion to the reader.

Response: Thank you very much for providing valuable feedback. We greatly appreciate your suggestions and have added a clear and concise description of our study design, specifying that it is a systematic review and meta-analysis of registered nurses' occupational fatigue. This includes the use of the Occupational Fatigue Exhaustion Recovery scale (OFER) to quantify fatigue levels and a systematic review of factors associated with nurses' occupational fatigue. Please refer to the revised version on page 6, lines 120-123.

Q5. Reporting bias assessment: The author should mention any procedures used to assess the likelihood of bias in a synthesis due to missing results (caused by reporting biases). Also, identify any methodologies used to determine certainty (or confidence) in the body of evidence supporting an outcome.

Response: Thank you for your valuable comments and suggestions on our manuscript. We have carefully addressed the concerns regarding the assessment of reporting bias and the certainty of the evidence supporting our outcomes. Studies with lower quality suggested a higher probability of bias, and we looked for any indications of unreported or selectively reported outcomes to minimize the impact of reporting bias. Please refer to the revised version on page 9, lines 165-169.

To ensure the certainty of our findings, we conducted a thorough assessment of the robustness of the meta-analysis results. Sensitivity analyses were performed to ascertain whether any of the studies in the meta-analysis produced changes in outcome. Additionally, we used Begg's tests to assess publication bias and conducted a visual inspection of funnel plots for asymmetry, which can be indicative of publication bias. Please refer to the revised version on page 10, lines 179-181, and page 10, lines 183-187.

Q6. RESULT: On study selection the authors should use the flow chart diagram.

Response: Thank you for your suggestion. We have used the flow chart diagram to show the literature screening process and results. Please refer to the Fig 1.

Q7. RESULT: The authors should arrange the section of the result well, as the same figure and table are not seen. Review and improve.

Response: We greatly appreciate your suggestion. We fully understand your concern and appreciate your efforts in reviewing our work. Regarding the figures, as per the submission guidelines, “Figure captions must be inserted in the text of the manuscript, immediately following the paragraph in which the figure is first cited (read order). Do not include captions as part of the figure files themselves or submit them in a separate document.” We have submitted them as individual files in the system.

As for the tables, we have placed each table directly after the paragraph in which it is first cited, maintaining the reading order. This should help readers to understand and analyze the data presented in the tables more effectively. Additionally, any supplementary tables (S1-S3 Tables) have been uploaded as supplementary materials for easy access.

Q8. Discussion: On part of limitations the authors should improve and make it clear.

Response: Thank you very much for reviewing our paper and providing valuable feedback. We fully understand and appreciate the points you raised regarding the “Limitations” section. We have elaborated on the limitations of the study in greater detail and clarity and provided directions for future research. We have discussed these limitations from three aspects: heterogeneity, study design, and sample selection. For each limitation, we have specified its manifestation and outlined potential areas for future research to address. Please refer to the revised version on page 36, lines 368-378.

Q9. Discuss implications of the results for practice, policy, and future research.

Response: We appreciate your constructive suggestions, which have been instrumental in guiding the improvement of my research paper. We have emphasized the importance of addressing cultural, organizational, and individual factors contributing to nurses' occupational fatigue and proposed specific strategies to mitigate its effects. We have also proposed further exploration into how these factors interplay in predicting fatigue and the development of innovative interventions to mitigate its effects in future research. Please refer to the revised version on page 35, lines 353-359, and page 36, lines 360-366.

Q10. References: Several references do not fit the requirements of Vancouver style. Revise and improve them.

Response: Thank you for your suggestion. We are sorry for our careless mistake. We have carefully reviewed and revised the references to ensure they meet the required format.

Replies to Reviewer #2

Reviewer #2:

Your study is very valuable, and I have some comments that may help improve your research:

Q1. The abstract does not clearly articulate the necessity of the work. Please make this more explicit.

Response: Thank you for your constructive comment. We have emphasized the pressing concern of occupational fatigue among nurses, highlighting its significant impact on productivity, error rates, and decision-making abilities. we have also clarified the urgent need for a comprehensive understanding of the global prevalence and multifaceted factors influencing nurses' occupational fatigue, given the scarcity of rigorous evaluations in the existing literature. Please refer to the revised version on page 2, lines 23-30.

Q2. Include a comprehensive list of your inclusion and exclusion criteria.

Response: Thank you for your valuable feedback. We apologize for not including a comprehensive list of my inclusion and exclusion criteria in the initial submission. To address this, we have incorporated a detailed table outlining these criteria below. You can also refer to the revised version on page 7, lines 140-142.

Table 1 Inclusion and exclusion criteria for the systematic review

Items Inclusion Exclusion

Study design Observational studies Qualitative studies/ experimental studies

Population Registered nurses Nurse managers, auxiliary nurses, and nurse educators

Instrument The Occupational Fatigue Exhaustion Recovery scale (OFER) ---

Outcomes The sample size, mean, and standard deviation of the scale, and factors associated with occupational fatigue insufficient data

Type of publication Peer reviewed full text papers Case reports, review articles, conference abstracts, comments, letters to the editor and protocols

Language English or Chinese Non-English and Non-Chinese

Q3. Specify the keywords used in your search and the types of studies that were included.

Response: Thank you for raising this important point. We have specified the keywords in our research. Please refer to the revised version on page 7, lines 132-134. Furthermore, in our inclusion and exclusion criteria for literature, we clearly specified the types of studies included. Specifically, we incorporated studies such as observational studies. Please refer to the revised version on page 7, lines 141-142.

Q4. Were there no studies published before 2009?

Response: Thank you for your careful review. In our literature search, we did consider relevant studies prior to 2009. However, after carefully evaluating the quality, relevance, and timeliness of these studies, we decided not to include them.

Q5. Indicate how many authors contributed to the search and quality assessment of the studies.

Response: Thank you for bringing this to our attention. We have added the important information. Please refer to the revised version on page 6, line 125 and page 9, lines 165-167.

Q6. In Table 1, please add the data collection instrument used for each study.

Response: Thank you for your suggestion to include the data collection instrument used for each study in Table 1. In our inclusion and exclusion criteria, we specifically noted that we included studies that utilized the Occupational Fatigue Exhaustion Recovery scale (OFER) for measurement. Given this clear focus on a single instrument, we initially decided not to redundantly list it in Table 1 for each study after further discussion, as it would be repetitive and potentially clutter the table. We acknowledge that this is a matter of style and preference and hope our response can address your concern.

Q7. At the end of the discussion, summarize the strengths of your study and provide suggestions for future research. Thank you for considering these points.

Response: Thank you very much for your valuable comments and suggestions on our research. We have added a summary of the strengths of this study at the end of the discussion section, emphasizing its comprehensiveness, methodological rigor, and consistency in using a sin

---

## [Decision Letter · Decision Letter 1]

Nurses' occupational fatigue level and risk factors: a Systematic Review and Meta- analysis

PONE-D-24-45547R1

Dear Dr. Li,

We’re pleased to inform you that your manuscript has been judged scientifically suitable for publication and will be formally accepted for publication once it meets all outstanding technical requirements.

Kind regards,

Fatma Refaat Ahmed, Ph.D.

Academic Editor

PLOS ONE

Additional Editor Comments (optional):

Reviewers' comments:

Reviewer's Responses to Questions

**Comments to the Author**

1. If the authors have adequately addressed your comments raised in a previous round of review and you feel that this manuscript is now acceptable for publication, you may indicate that here to bypass the “Comments to the Author” section, enter your conflict of interest statement in the “Confidential to Editor” section, and submit your "Accept" recommendation.

Reviewer #1: All comments have been addressed

Reviewer #3: All comments have been addressed

2. Is the manuscript technically sound, and do the data support the conclusions?

Reviewer #1: Yes

Reviewer #3: Yes

3. Has the statistical analysis been performed appropriately and rigorously? 

Reviewer #1: Yes

Reviewer #3: Yes

4. Have the authors made all data underlying the findings in their manuscript fully available?

Reviewer #1: Yes

Reviewer #3: Yes

5. Is the manuscript presented in an intelligible fashion and written in standard English?

Reviewer #1: Yes

Reviewer #3: Yes

6. Review Comments to the Author

Reviewer #1: REVIEWER COMMENT AND SUGGESTIONS

I was able to examine closely and realize that the author has been able to give paths for each step of writing the procedures in manuscript; however there are a few things authors need to make it clear. On part of discussion and limitation to the first paragraph the authors should revise and use correct English used in research. Also I noted line no 201,227,228,229 FIGURE are not seen.

Thank you

Reviewer #3: Congratulations for this valuable paper. The comments have been addressed thoroughly and effectively incorporated into the manuscript.

7. PLOS authors have the option to publish the peer review history of their article (what does this mean? ). If published, this will include your full peer review and any attached files.

**Do you want your identity to be public for this peer review?** For information about this choice, including consent withdrawal, please see our Privacy Policy .

Reviewer #1: **Yes: ** rehema abdallah

Reviewer #3: No

---

## [Editor Report · Acceptance letter]

PONE-D-24-45547R1

PLOS ONE

Dear Dr. Li,

I'm pleased to inform you that your manuscript has been deemed suitable for publication in PLOS ONE. Congratulations! Your manuscript is now being handed over to our production team.

Kind regards,

on behalf of

Dr. Fatma Refaat Ahmed

Academic Editor

PLOS ONE